# MR Relaxometry for Discriminating Malignant Ovarian Cystic Tumors: A Prospective Multicenter Cohort Study

**DOI:** 10.3390/diagnostics14111069

**Published:** 2024-05-21

**Authors:** Naoki Kawahara, Hiroshi Kobayashi, Tomoka Maehana, Kana Iwai, Yuki Yamada, Ryuji Kawaguchi, Junko Takahama, Nagaaki Marugami, Hirotaka Nishi, Yosuke Sakai, Hirokuni Takano, Toshiyuki Seki, Kota Yokosu, Yukihiro Hirata, Koyo Yoshida, Takafumi Ujihira, Fuminori Kimura

**Affiliations:** 1Department of Obstetrics and Gynecology, Nara Medical University, Kashihara 634-8522, Japan; hirokoba@naramed-u.ac.jp (H.K.); tmaehana@naramed-u.ac.jp (T.M.); iwaikana@naramed-u.ac.jp (K.I.); yuki0528@naramed-u.ac.jp (Y.Y.); kawaryu@naramed-u.ac.jp (R.K.); kimurafu@naramed-u.ac.jp (F.K.); 2Department of Gynecology and Reproductive Medicine, Ms. Clinic MayOne, 871-1 Shijo-Cho, Kashihara 634-0813, Japan; 3Department of Radiology, Higashiosaka City Medical Center, Higashiosaka 578-8588, Japan; takahama-j@higashiosaka-hosp.jp; 4Department of Radiology and Nuclear Medicine, Nara Medical University, Kashihara 634-8522, Japan; marugami@naramed-u.ac.jp; 5Department of Obstetrics and Gynecology, Tokyo Medical University, Shinjuku-Ku, Tokyo 160-0023, Japan; nishih@tokyo-med.ac.jp (H.N.); sakai-y@tokyo-med.ac.jp (Y.S.); 6Department of Obstetrics and Gynecology, The Jikei University Kashiwa Hospital, Kashiwa 277-8567, Japan; hirokuni@jikei.ac.jp (H.T.); tseki015@gmail.com (T.S.); h18ms-yokosu@jikei.ac.jp (K.Y.); 7Department of Obstetrics and Gynecology, The Jikei University School of Medicine, Minato-Ku, Tokyo 105-8461, Japan; yukihiro03121982@yahoo.co.jp; 8Department of Obstetrics and Gynecology, Juntendo University Urayasu Hospital, Urayasu 279-0021, Japan; koyo.yoshida@juntendo-urayasu.jp (K.Y.); tujihira@juntendo.ac.jp (T.U.)

**Keywords:** endometriosis-associated ovarian cancer (EAOC), ovarian endometrioma (OE), magnetic resonance (MR) relaxometry, prospective multicenter cohort study

## Abstract

Background: Endometriosis-associated ovarian cancer (EAOC) is a well-known type of cancer that arises from ovarian endometrioma (OE). OE contains iron-rich fluid in its cysts due to repeated hemorrhages in the ovaries. However, distinguishing between benign and malignant tumors can be challenging. We conducted a retrospective study on magnetic resonance (MR) relaxometry of cyst fluid to distinguish EAOC from OE and reported that this method showed good accuracy. The purpose of this study is to evaluate the accuracy of a non-invasive method in re-evaluating pre-surgical diagnosis of malignancy by a prospective multicenter cohort study. Methods: After the standard diagnosis process, the R2 values were obtained using a 3T system. Data on the patients were then collected through the Case Report Form (CRF). Between December 2018 and March 2023, six hospitals enrolled 109 patients. Out of these, 81 patients met the criteria required for the study. Results: The R2 values calculated using MR relaxometry showed good discriminating ability with a cut-off of 15.74 (sensitivity 80.6%, specificity 75.0%, AUC = 0.750, *p* < 0.001) when considering atypical or borderline tumors as EAOC. When atypical and borderline cases were grouped as OE, EAOC could be distinguished with a cut-off of 16.87 (sensitivity 87.0%, specificity 61.1%). Conclusions: MR relaxometry has proven to be an effective tool for discriminating EAOC from OE. Regular use of this method is expected to provide significant insights for clinical practice.

## 1. Introduction

Ovarian cancer is a leading cause of death in women, ranking fifth among all cancer-related deaths [1]. Unfortunately, this disease is often referred to as a “silent killer” [2,3,4] because it cannot be detected in its early stages. As a result, most cases of ovarian cancer are diagnosed when the disease has already progressed to advanced stages [5,6,7,8], and over 185,000 people worldwide die from this disease each year [9,10].

Epithelial ovarian cancer can be classified into two distinct categories, types 1 and 2 [11,12,13,14,15], based on their molecular and morphological features. Type 1 tumors progress slowly through a sequence known as adenoma-carcinoma. This includes several types of ovarian cancer, such as endometriosis-associated ovarian cancer (EAOC), mucinous carcinoma, low-grade serous carcinoma, clear cell carcinoma (CCC), and low-grade endometrioid carcinoma [16]. On the other hand, type 2 tumors progress rapidly, such as high-grade serous and endometrioid carcinoma, malignant mixed mesodermal tumors (carcinosarcomas), and undifferentiated carcinoma [17,18,19]. Type 1 tumors have been reported to give rise to EAOC from ovarian endometrioma (OE), which is characterized by endometriotic glands and hemosiderosis in the stromal area. OE is characterized by repeated hemorrhages in the ovaries, which may result in several symptoms such as dysmenorrhea [20,21,22], chronic pelvic pain [23,24,25,26], and infertility [27,28,29,30], as well as the genesis of EAOC at a rate of 0.72% [31]. EAOC, which arises from OE, contains an iron-rich fluid in its cysts [32].

Mural nodules and papillary projections are often considered indicators of cancer [33]. In general, the existence of solid parts in ovarian cancer that display high signal intensity on diffusion images and low apparent diffusion coefficient (ADC) values is indispensable [34]. It is well known that the magnetic resonance imaging (MRI) findings of EC, CCC, and HGSC often show similar large cystic masses with hyperintensity on T1- and T2-weighted imaging (T1/T2WI) and intra-cystic vascularized mural nodules and/or solid components [35,36,37]. Recent studies have shown that the growth pattern of mural nodules and the height-to-width ratio (HWR) can discriminate between CCC and EC [38] and whole-tumor ADC histogram analysis among SMBT, CCC, and EC [39]. However, these indicators can also be present in non-cancerous OE, which can make it difficult for doctors to diagnose before surgery [40,41,42,43]. For instance, ADC values have been observed to differ based on the subtype of epithelial ovarian carcinoma [44], which can overlap with those of borderline tumors, making it difficult to arrive at an accurate diagnosis [45]. We reported that magnetic resonance (MR) relaxometry can distinguish between EAOC and OE by measuring the iron concentration in the cyst fluid in a non-invasive manner [46], which allows for differentiating between EAOC and OE with high accuracy [47]. Because of the nature of the retrospective study, there could be bias, and prospective studies are needed to demonstrate that OE and EAOC can be accurately differentiated. 

The main purpose of this study is to assess how MR relaxometry can distinguish between OE and EAOC in a prospective multicenter cohort setting. This will enable future comparisons with conventional methods, such as dynamic contrast enhancement, diffusion, and ADC values. As a secondary goal, the study will also assess the ability to distinguish between HGSC (as non-EAOC), CCC and EC (as EAOC), and OE and serous cyst adenoma (benign tumor).

## 2. Materials and Methods

### 2.1. Patients

The effect size in the previous study was 0.713 [47], setting a significance level of 0.01 and a power of 0.95; the minimum number of cases required was 19, and the target number of cases in each group was set at 30. This is a study that involves multiple medical centers and aims to compare preoperative R2 values of patients who are suspected of having endometriosis-related ovarian cancer, non-endometriosis-related ovarian cancer, endometriomas, and serous cysts with postoperative pathological and other examination results. The primary endpoint of this study is to differentiate between OE and EAOC by R2 values, while the secondary endpoint is to differentiate between non-EAOC and EAOC or serous cyst. A cohort of chemotherapy-naive patients with histologically confirmed primary ovarian tumors was recruited at six hospitals between December 2018 and March 2023. The study was conducted in accordance with the ethical principles outlined in the Declaration of Helsinki, the International Council for Harmonization Guideline for Good Clinical Practice, and all applicable local regulatory requirements. The Institutional Review Board approved the study, including the protocol and informed consent form (Approval Number: 2021). Before undertaking any study-related procedures, informed consent was obtained from each patient. In case of study protocol modification, the information was notified and confirmed after approval by the review board and the University Hospital Medical Information (UMIN) Clinical Trials Registry (UMIN-CTR) using UMIN000034969. Inclusion was allowed for patients who were over 20 years old and had epithelial ovarian tumors. Patients with metastatic ovarian tumors, tumors less than 5 cm in diameter, distinctive mucinous cyst adenoma or dermoid cyst, those who were unable to undergo MRI due to claustrophobia, patients with mental manifestations or psychiatric disorders, those who were unable to indicate their intention, and those deemed inadequate by researchers were excluded from the study. These tumors were excluded due to difficulties in determining their position or if they were non-primary epithelial tumors of the ovary. A total of 109 patients were included in the current cohort. The flow chart of the current cohort is outlined in Figure 1. 

The planned number of OE and EAOC cases was reached, but non-EAOC (high-grade serous carcinoma (HGSC)) and serous cysts were not. Patients with OE mainly received laparoscopic surgery, and the patients suspected of harboring malignant tumors underwent laparotomy. The following factors were gathered from the Case Report Form (CRF): age, body mass index (BMI), gravida, parity, postoperative diagnosis (including FIGO 2014 stage) with TNM classification (UICC 8th), tumor diameter, tumor volume, pre-treatment blood test results (such as carbohydrate antigen125 [CA125], carbohydrate antigen 19-9 [CA 19-9]), and R2 value.

### 2.2. Tumor Imaging and Diagnoses

All patients first visited the outpatient clinic, where they underwent an internal examination that included ultrasound. After ultrasound, routine MR imaging was performed using T1W and T2W sequences. The tumor diameter was recorded as the largest diameter found among the axial, sagittal, and coronal imaging (D1, D2, and D3). From the three diameters, tumor volume was calculated using following the formula: D1 × D2 × D3 × π/6. After undergoing routine clinical MR imaging, the registered patients underwent MR relaxometry using a 3T system (Skyla, Siemens Healthcare, Erlangen, Germany). An exponential decay was fitted to the echo amplitude at multiple echo times [48]. A parameter R2 value (s^−1^) was then calculated using a high-speed T2*-corrected multi-echo MR sequence (HISTO) by the 3T MR system in vivo. This sequence allows for the estimation of liver iron deposition, since the T2 of water changes with iron concentration. The pulse sequence design and programming were done using an imaging platform from Siemens Medical Systems, Erlangen, Germany, and applied to the 3T system. The HISTO sequence was based on single voxel STEAM sequences that could be used for relative fat quantification in the liver. The sequence consisted of five measurements with different echo times (TEs), namely 12, 24, 36, 48, and 72 ms. The typical protocol involved holding one’s breath and taking a total of 15 s to acquire the data. The repetition time (TR) was set to 3000 ms to compensate for the signal saturation effects while maintaining a reasonable acquisition time. A 15 × 15 × 15 mm spectroscopy voxel (VOI) was used to select a region that included only the liquid portion of the cyst and not the solid portion. If a patient had more than one cyst, the fluid from the largest cyst was measured. The VOI was placed at the center of the OE or EAOC cyst by a radiologist who specializes in female pelvic MR imaging. After MR relaxometry, four patients were diagnosed with endometrioma by imaging and followed up as outpatients. All the other cases were diagnosed using surgically removed tissue. The diagnosis was confirmed by histological examination of the surgically removed tissue. At least two pathologists who were blinded to the study conducted the histological examination.

### 2.3. Statistical Analysis

Analyses were conducted using IBM SPSS version 25.0. A Shapiro–Wilk test was performed before each test to ensure that non-normal distributions were included. To compare the differences of each factor among the groups, a Mann–Whitney U test, Kruskal–Wallis one-way ANOVA test, or chi-square test was used. To predict malignant ovarian tumors, a receiver operating characteristic (ROC) curve analysis was performed to determine the cut-off value. The cut-off value was based on the highest Youden index, which is calculated as sensitivity plus specificity minus one. A *p*-value of less than 0.05 was considered to indicate a statistically significant difference.

## 3. Results

### 3.1. Patients and Parameters to Discriminate Each Tumor Subtype

Between December 2018 and March 2023, a total of 109 patients were enrolled in this study. Out of these, 28 were excluded because they had non-targeted tumor subtypes such as mucinous cystadenoma or dermoid tumor. The remaining patients were divided into different categories based on their tumor subtypes: 31 had EAOC, 7 had non-EAOC, 28 had OE, and 15 had a serous cyst. The demographic and clinical characteristics of all the patients in the combined cohort are shown in Table 1.

Table 2 shows the results of the pre-surgery examinations in the current cohort. 

When distinguishing between EAOC and non-EAOC, only CA 19-9 was significant (*p* < 0.001). To differentiate between non-EAOC and serous cyst, CA125 and the largest tumor diameter were significant factors (*p* = 0.002 and *p* = 0.021). When distinguishing EAOC from OE, age (*p* < 0.001), CA125 (*p* = 0.047), CA 19-9 (*p* = 0.001), largest tumor diameter (*p* < 0.001), tumor volume (*p* < 0.001), and R2 value (*p* = 0.001) were all reliable indicators for discrimination.

### 3.2. Primary Endpoint: The Discrimination of EAOC from OE

Table 3 displays the results of the ROC curve analysis for detecting malignant tumors.

The optimal cut-off value was determined by analyzing the ROC curve between EAOC and OE. When considering atypical or borderline tumors as EAOC, OE can be distinguished with a cut-off of 15.74. The sensitivity and specificity of this cut-off are 80.6% and 75.0%, respectively, with an AUC of 0.750 (*p* < 0.001) (Figure 2a). EAOC can be distinguished with a cut-off of 16.87, separating atypical/borderline cases from EAOC ones (sensitivity 87.0%, specificity 61.1%, AUC = 0.750, *p* < 0.001) (Figure 2b).

### 3.3. Secondary Endpoint: Discriminating between EAOC and Non-EAOC or between Non-EAOC and Serous Cyst

When differentiating between EAOC and non-EAOC, only CA 19-9 demonstrated significance with a cut-off value of 21 U/mL. This yielded a sensitivity of 80.6%, a specificity of 100%, and an AUC of 0.919 with a *p*-value of 0.001. On the other hand, to distinguish non-EAOC from serous cysts, CA125, and the largest tumor diameter emerged as significant factors. A cut-off value of 74 U/mL for CA125 gave a sensitivity of 71.4%, specificity of 100%, and AUC of 0.896, with a *p*-value of 0.004. Similarly, a cut-off value of 102 mm for the largest tumor diameter gave a sensitivity of 71.4%, specificity of 93.3%, and AUC of 0.810, with a *p*-value of 0.022.

## 4. Discussion

In the multicenter prospective study, the effectiveness of MR relaxometry to differentiate EAOC from OE was demonstrated for the first time in the world. In a previous report, we found that MR relaxometry could serve as a noninvasive tool to predict malignant transformations before surgery. We previously reported a favorable predictive accuracy in the investigative study, with a sensitivity of 86% and specificity of 94% [47]. However, in the current study, the accuracy was lower than in the previous report, which could be attributed to the nature of the multicenter prospective study. Actually, this study included eight cases of borderline or atypical tumors. When considering atypical or borderline tumors as either OE or EAOC, the cut-off value of R2 changes slightly. This is because the R2 value of the cyst fluids of atypical or borderline tumors is intermediate, as shown in Appendix A. This finding can aid in the stepwise hypothesis of OE to EAOC [41], which suggests that the reduction of radical species related to iron components plays a critical role in the carcinogenesis of OE. Ovarian atypical endometrioma or borderline tumors are premalignant lesions that are difficult to diagnose before surgery. For instance, ovarian atypical endometrioma cannot be distinguished from EAOC by pre-surgical CA 125 or HE4 [49], but it can be differentiated by age. Borderline tumors that arise from OE are also more difficult to diagnose because of intra-cystic vascularized mural nodules, and they exhibit similar characteristics on MRI imaging [41]. This study provided an essential clue not only to accurate diagnosis in situations where physicians face challenges determining whether to conduct a surgical resection but also to prove the above hypothesis.

Some methods have been used to discriminate malignant ovarian tumors from benign tumors. We previously developed a new tool called the Endometriotic Neoplasm Algorithm for Risk Assessment (e-NARA) index. This index considers key factors, such as age, the longest tumor diameter, and R2 value, which significantly improves its accuracy in distinguishing EAOC from OE. As a result, the index has a high sensitivity of 85.7% and a specificity of 87.0% [50]. In addition, transvaginal ultrasound is an effective and cost-efficient method for detecting malignant transformation of ovarian endometrioma. It is easily operable and can be performed in an outpatient clinic. The International Ovarian Tumor Analysis (IOTA) group has developed a highly accurate classification system [51]. Lee Cohen Ben-Meir et al. investigated this method concerning ovarian endometriosis (OE) and its associated malignant tumor, endometrioid, and clear cell ovarian carcinoma (EAOC). They reported that this method can distinguish malignant tumors with high sensitivity [52]. In clinical settings, distinguishing atypical or borderline tumors from OE or advanced EAOC and deciding on the appropriate surgical method can be a challenging and important process for physicians. The above methods could help guide decisions in such cases.

This study also provides important evidence to discriminate tumors among EAOC, non-EAOC, and serous cysts. CA 19-9 is typically found to be elevated in tumors affecting the gastrointestinal tract, biliary tract, or ovaries. Santotoribio et al. reported that the sensitivity and specificity of CA 19-9 in screening for ovarian cancer were 50% and 97.6%, respectively [53]. According to another study, the combination of serum CA19-9 and HE4 can be useful in diagnosing EAOC from OE [54]. While our study is the first to show that CA 19-9 can accurately distinguish EAOC from non-EAOC, it should be noted that the limited number of non-EAOC cases calls for caution in concluding that CA19-9 is effective in discriminating between these subtypes. Figure 3 illustrates the ovarian tumor discrimination landscape based on the current multicenter cohort study. 

This study not only provides a crucial key to detecting malignant transformation from OE but also elucidates tumor subtypes just by pre-surgical clues, which can shed light on omitting invasive surgical biopsy and enable immediate intervention.

## 5. Conclusions

MR relaxometry has proved to be an effective tool for discriminating EAOC from OE. The regular use of this method is expected to provide significant insights for clinical practice.

## Figures and Tables

**Figure 1 diagnostics-14-01069-f001:**
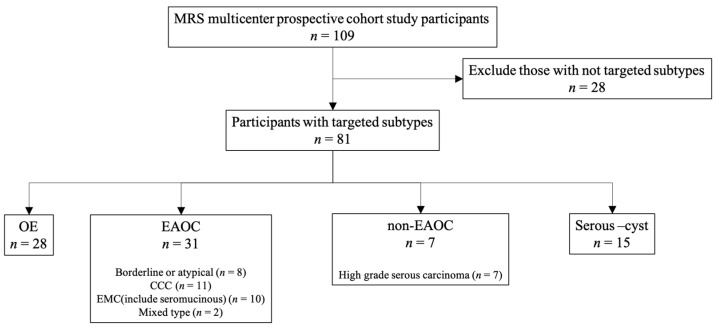
The flow chart of the current cohort. A total of 81 patients were enrolled in the study after the exclusion of 28 patients because of non-targeted tumor subtypes such as mucinous cystadenoma or dermoid tumor. OE, ovarian endometrioma; EAOC, endometriosis-associated ovarian cancer; MRS, MR spectroscopy; CCC, clear cell carcinoma; EMC, endometrioid carcinoma.

**Figure 2 diagnostics-14-01069-f002:**
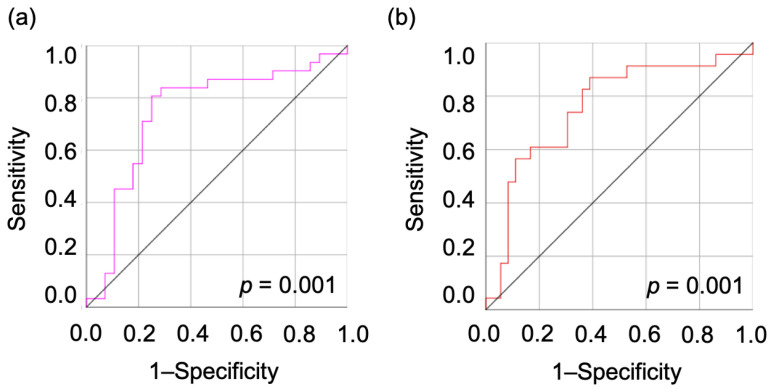
The ROC curve distinguishes OE from EAOC with a cut-off of 15.74 including atypical/borderline cases as EAOC (**a**), and with a cut-off of 16.87 excluding atypical/borderline cases from EAOC ones (**b**). ROC, receiver operating characteristic; OE, ovarian endometrioma; EAOC, endometriosis-associated ovarian cancer.

**Figure 3 diagnostics-14-01069-f003:**
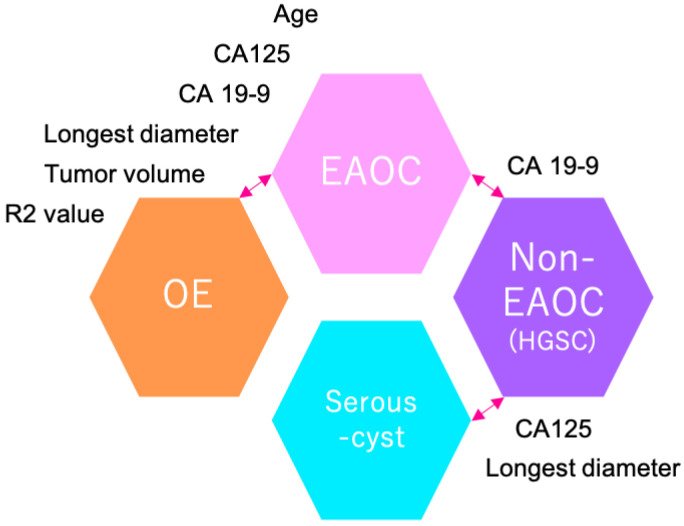
The landscape for ovarian tumor discrimination based on the current multicenter cohort study. OE, ovarian endometrioma; EAOC, endometriosis-associated ovarian cancer; CA125, carbohydrate antigen 125; CA 19-9, carbohydrate antigen 19-9.

**Table 1 diagnostics-14-01069-t001:** Demographic and clinical characteristics of the current cohort.

	EAOC	Non-EAOC	OE	Serous-Cyst	*p*-Value
Number	*n* = 31	*n* = 7	*n* = 28	*n* = 15	
Age (years)					
Median (range)	53 (33–78)	52 (46–64)	37 (24–80)	61 (21–79)	
Mean ± SD	54.13 ± 13.00	53.86 ± 6.96	38.28 ± 10.84	59.13 ± 16.31	<0.001
BMI					
Median (range)	22.41 (17.23–39.48)	20.52 (19.63–26.35)	21.68 (14.88–40.09)	23.32 (17.87–36.20)	
Mean ± SD	23.41 ± 5.11	21.45 ± 2.40	22.68 ± 6.28	23.61 ± 4.74	0.534
Gravida					
0	14	1	13	3	
≥1	17	6	16	12	0.190
Parity					
0	15	1	14	4	
≥1	16	6	15	11	0.212

EAOC, endometriosis-associated ovarian cancer; OE, ovarian endometrioma; BMI, body mass index. Significant differences were found between groups in age factor, with the lowest in OE patients (*p* < 0.001).

**Table 2 diagnostics-14-01069-t002:** Results of pre-treatment examinations in the current cohort.

	EAOC	Non-EAOC	OE	Serous-Cyst	*p*-Value
Number	*n* = 31	*n* = 7	*n* = 29	*n* = 15	
CA125 (U/mL)			*^1^	*^2^	
Median (range)	113.0 (12.0–14.0 × 10^3^)	102.0 (13.4–617.0)	64.5 (9.8–251.0)	12.0 (6.0–57.0)	
Mean ± SD	1.34 × 10^3^ ± 3.12 × 10^3^	185.91 ± 219.33	74.35 ± 56.71	17.84 ± 16.77	<0.001
CA19-9 (U/mL)			*^3^	*^2^	
Median (range)	64.0 (2.0–199.4 × 10^3^)	7.0 (3.0–17.5)	20.0 (1.0–57.0)	8.0 (2.0–53.0)	
Mean ± SD	7.21 × 10^3^ ± 35.74 × 10^3^	8.31 ± 4.56	23.03 ± 15.29	12.76 ± 13.66	<0.001
Longest diameter (mm)					
Median (range)	116.00 (57.00–202.00)	107.90 (72.64–161.21)	73.60 (23.53–130.00)	75.41 (60.13–122.80)	
Mean ± SD	123.15 ± 45.41	113.69 ± 35.37	72.08 ± 26.62	80.14 ± 16.43	<0.001
Tumor volume (cm^3^)					
Median (range)	499.07 (85.35–2.69 × 10^3^)	271.14 (131.13–818.90)	119.09 (5.36–637.94)	162.21 (87.60–521.20)	
Mean ± SD	750.02 ± 729.35	363.70 ± 253.89	172.72 ± 162.96	211.74 ± 125.11	<0.001
R2 value (s^−1^)					
Median (range)	10.89 (4.50–63.00)	9.52 (5.01–12.00)	21.55 (4.53–59.42)	9.68 (7.20–13.28)	
Mean ± SD	14.38 ± 12.33	9.09 ± 2.21	23.40 ± 13.67	9.99 ± 1.51	<0.001

EAOC, endometriosis-associated ovarian cancer; OE, ovarian endometrioma; CA125, carbohydrate antigen 125; CA 19-9, carbohydrate antigen 19-9. *^1^ One item missing in data; *^2^ two items missing in data; *^3^ four items missing in data.

**Table 3 diagnostics-14-01069-t003:** The cut-off values for the differentiation of EAOC from OE.

	AUC	*p*-Value	Cut-Off Value	Sensitivity	Specificity
Age (year)	0.838	<0.001	48	0.677	0.931
CA125 (U/mL)	0.651	0.047	115.5	0.484	0.929
CA19-9 (U/mL)	0.752	0.001	60.5	0.516	1.000
Longest diameter (mm)	0.813	<0.001	85.7	0.742	0.759
Tumor volume (cm^3^)	0.800	<0.001	490.5	0.516	0.966
R2 value (s^−1^)	0.750	0.001	15.7	0.806	0.759

CA125, carbohydrate antigen 125; CA 19-9, carbohydrate antigen 19-9; AUC, area under curve.

## Data Availability

Data are contained within the article.

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
