# Peer review of "MR Relaxometry for Discriminating Malignant Ovarian Cystic Tumors: A Prospective Multicenter Cohort Study"

_diagnostics, 2024, doi:10.3390/diagnostics14111069_

Round 1
Reviewer 1 Report
Comments and Suggestions for Authors
This is an interesting study aiming to assess the role of Magnetic Resonance relaxometry in discriminating endometriosis-associated ovarian cancer (EAOC) from ovarian endometrioma (OE). It is a prospective multicenter cohort study involving 81 participants.
The study used the R2 value calculated by the MR relaxometry and using a cut-off of around 16 obtained sensitivity 80.6%, specificity 75.0% with a very good outcome for AUC = 0.750 (p < 0.001) when atypical or borderline cases were grouped with EAOC. In a second scenario, atypical and borderline cases were grouped as OE, then with a cut-of around 17 the sensitivity was 87.0% and specificity 61.1%. The authors have also evaluated other classical “quantities” as discriminators.
The manuscript is in general well written and presented. There are no severe linguistic issues and, in my opinion, a few minor corrections may be required. Follows a list of comments that may help to improve the manuscript.
1. Line 90: “The effect size in the previous study was 0.713” please add a bibliography of the study here
2. “Inclusion was allowed for patients who were over 20 years old and had epithelial ovarian tumors” Please clarify why not 18 years since this is the standard?
3. Figure 1: an arrow is almost invisible, improvement in resolution may be required
4. Line 158: “Analyses were conducted using IBM SPSS version 25.0 in Armonk, NY, USA” there is something wrong in syntax here
5. In the statistical analysis paragraph “a Mann-Whitney U test, Kruskal-Wallis one-way ANOVA test, or chi-square test was used “. The authors have applied non parametric tests, however when normality is ensured is better to apply parametric tests, thus please clarify if there are performed tests for normality to decide for non-parametric tests.
Comments on the Quality of English LanguageQuality of language is OK, minor syntax issues were detected.
Author Response
Dear reviewer,
The following are our point-by-point replies. Please check it.
This is an interesting study aiming to assess the role of Magnetic Resonance relaxometry in discriminating endometriosis-associated ovarian cancer (EAOC) from ovarian endometrioma (OE). It is a prospective multicenter cohort study involving 81 participants.
The study used the R2 value calculated by the MR relaxometry and using a cut-off of around 16 obtained sensitivity 80.6%, specificity 75.0% with a very good outcome for AUC = 0.750 (p < 0.001) when atypical or borderline cases were grouped with EAOC. In a second scenario, atypical and borderline cases were grouped as OE, then with a cut-of around 17 the sensitivity was 87.0% and specificity 61.1%. The authors have also evaluated other classical “quantities” as discriminators.
The manuscript is in general well written and presented. There are no severe linguistic issues and, in my opinion, a few minor corrections may be required. Follows a list of comments that may help to improve the manuscript.
- Line 90: “The effect size in the previous study was 0.713” please add a bibliography of the study here
-I agree. I added the bibliography.
- “Inclusion was allowed for patients who were over 20 years old and had epithelial ovarian tumors” Please clarify why not 18 years since this is the standard?
-This setup method is standard in Japan.
- Figure 1: an arrow is almost invisible, improvement in resolution may be required
-Because the resolution was enough as 300 dpi, I enlarged the arrowhead.
- Line 158: “Analyses were conducted using IBM SPSS version 25.0 in Armonk, NY, USA” there is something wrong in syntax here
-You are right. Thank you very much.
- In the statistical analysis paragraph “a Mann-Whitney U test, Kruskal-Wallis one-way ANOVA test, or chi-square test was used “. The authors have applied non parametric tests, however when normality is ensured is better to apply parametric tests, thus please clarify if there are performed tests for normality to decide for non-parametric tests.
-I agree. We performed the Shapiro-Wilk test before each test and then ensured that a non-normal distribution was included.
Reviewer 2 Report
Comments and Suggestions for Authors
Dear colleagues,
I read your manuscript and find it very interesting. You could introduce a non invasive method for distinguishing the ovaries benign tumors from the malignant masses.
I could not find the base of your comparisons and evaluations. You announced you had 81 patients, ok, did they have any pathology to find their exact diagnoses? How could you be sure about 15 Serous-cyst and 29 OE for example? You did MR relaxometry after being sure about the lesions pathology or before that?
Please re write your materials and methods and mention this important point.
I am waiting to hear from you soon.
Author Response
Dear reviewer,
The following are our point-by-point replies. Please check it.
I read your manuscript and find it very interesting. You could introduce a non invasive method for distinguishing the ovaries benign tumors from the malignant masses.
I could not find the base of your comparisons and evaluations. You announced you had 81 patients, ok, did they have any pathology to find their exact diagnoses? How could you be sure about 15 Serous-cyst and 29 OE for example? You did MR relaxometry after being sure about the lesions pathology or before that?
Please re write your materials and methods and mention this important point.
I am waiting to hear from you soon.
-Thank you very much for pointing this out. I modified this section more precisely and chronologically.